# Learning a High Fidelity Pose Invariant Model for High-resolution Face Frontalization

**Jie Cao, Yibo Hu, Hongwen Zhang, Ran He,\* Zhenan Sun**
National Laboratory of Pattern Recognition, CASIA
Center for Research on Intelligent Perception and Computing, CASIA
Center for Excellence in Brain Science and Intelligence Technology, CASIA
University of Chinese Academy of Sciences, Beijing, 100049, China
{jie.cao,yibo.hu,hongwen.zhang}@cripac.ia.ac.cn  {rhe,znsun}@nlpr.ia.ac.cn

## Abstract

Face frontalization refers to the process of synthesizing the frontal view of a face from a given profile. Due to self-occlusion and appearance distortion in the wild, it is extremely challenging to recover faithful results and preserve texture details in a high-resolution. This paper proposes a High Fidelity Pose Invariant Model (HF-PIM) to produce photographic and identity-preserving results. HF-PIM frontalizes the profiles through a novel texture warping procedure and leverages a dense correspondence field to bind the 2D and 3D surface spaces. We decompose the prerequisite of warping into dense correspondence field estimation and facial texture map recovering, which are both well addressed by deep networks. Different from those reconstruction methods relying on 3D data, we also propose Adversarial Residual Dictionary Learning (ARDL) to supervise facial texture map recovering with only monocular images. Exhaustive experiments on both controlled and uncontrolled environments demonstrate that the proposed method not only boosts the performance of pose-invariant face recognition but also dramatically improves high-resolution frontalization appearances.

## 1 Introduction

Face frontalization refers to predicting the frontal view image from a given profile. It is an effective preprocessing method for pose-invariant face recognition. Frontalized profile faces can be directly used by general face recognition methods without retraining the recognition models. Recent studies have shown that frontalization is a promising approach to address long-standing problems caused by pose variation in face recognition system. Additionally, generating photographic frontal faces are beneficial for a series of face-related tasks, including face reconstruction, face attribute analysis, facial animation, etc.

Due to the appealing prospect in theories and applications, research interest has been lasting for years. In the early stage, most traditional face frontalization methods [8, 14, 15, 9, 40] are 3D-based. These methods mainly leverage theories in monocular face reconstruction to recover 3D faces, and then render frontal view images. The well-known 3D Morphable Model (3DMM) [2] has been widely employed to express facial shape and appearance information. Recently, great breakthroughs have been made by the methods based on generative adversarial networks (GAN) [10]. Those methods frontalize faces from the perspective of 2D image-to-image translation and build deep networks with novel architectures. The visual realism has been improved significantly, for instance, in Multi-PIE [11], some synthesized results [20, 37] from small pose profiles are so photographic that it is difficult for human observers to distinguish them from the real ones. Furthermore, frontalized

results have been proved to be effective to tackle the pose discrepancy in face recognition. Through the "recognition via generation" framework, i.e., rotating the profiles to the frontal views, which can be directly used by general face recognition methods, frontalization methods [36, 37] achieve state-of-the-art pose-invariant face recognition performance on multiple datasets, including Multi-PIE and IJB-A [23].

Even though much progress has been made, there are still some ongoing issues for in-the-wild face frontalization. For traditional 3D-based approaches, due to the shortage of 3D data and the limited representation power of backbone 3D model, their performances are commonly less competitive compared with GAN-based methods albeit some improvements [5, 32] have been made. However, GAN-based methods heavily rely on minimizing pixel-wise losses to deal with the noisy data for in the wild settings. As discussed in many other image restoration tasks [19, 21], the consequence is that the outputs lack variations and tend to keep close to the statistical meaning of the training data. The results will be over-smoothed with little high-level texture information. Hence, current frontalization results are less appealing in a high-resolution and the output size is often no larger than $128 \times 128$.

To address the above issues, this paper proposes a High Fidelity Pose Invariant Model (HF-PIM) that combines the advantages of 3D and GAN based methods. In HF-PIM, we frontalize the profiles via a novel texture warping procedure. Inspired by recent progress in 3D face analysis [12, 13], we introduce a dense correspondence field to bind the 2D and 3D surface spaces. Thus, the prerequisite of our warping procedure is decomposed into two well-constrained problems: dense correspondence field estimation and facial texture map recovering. We build a deep network to address the two problems and benefit from its greater representation power than traditional 3D-based methods. Furthermore, we propose Adversarial Residual Dictionary Learning (ARDL) to get rid of the heavy reliance on 3D data. Thanks to the 3D-based deep framework and the capacity of ARDL for fine-grained texture representation [6], high-resolution results with faithful texture details can be obtained. We make extensive comparisons with state-of-the-art methods on the IJB-A, LFW [18] and Multi-PIE datasets. We also frontalize $256 \times 256$ images from CelebA-HQ [22] to push forward the advance in high-resolution face frontalization. Quantitative and qualitative results demonstrate our HF-PIM dramatically improves pose-invariant face recognition and produces photographic high-resolution results potentially benefitting many real-world applications.

To summarize, our main contributions are listed as follows:

- A novel High Fidelity Pose Invariant Model (HF-PIM) is proposed to produce more realistic and identity-preserving frontalized face images with a higher resolution.

- Through dense correspondence field estimation and facial texture map recovering, our warping procedure can frontalize profile images with large poses and preserves abundant latent 3D shape information.

- Without the need of 3D data, we propose ARDL to supervise the process of facial texture map recovering, effectively compensating the texture representation capacity for 3D-based framework.

- A unified end-to-end deep network is built to integrate all algorithmic components, which makes the training process elegant and flexible.

- Extensive experiments on four face frontalization databases demonstrate that HF-PIM not only boosts pose-invariant face recognition in the wild, but also dramatically improves the visual quality of high-resolution images.

## 2   Related Works

In recent years, GAN, proposed by Goodfellow et al. [10], has been successfully introduced into the field of computer vision. GAN can be regarded as a two-player non-cooperative game model. The main components, generator and discriminator, are rivals of each other. The generator tries to map a given input distribution to a target data distribution. Whereas the discriminator tries to distinguish the data produced by the generator from the real one. Recently, deep convolutional generative adversarial network (DCGAN) [29] has demonstrated the superior performance of image generation. Info-GAN [4] applies information regularization to optimization. Furthermore, Wasserstein GAN [1] improves the learning stability of GAN and provides solutions for debugging and hyperparameter searching

for GAN. These successful theoretical analyses of GAN show the effectiveness and possibility of photorealistic face image generation and synthesis.

GAN has dominated the field of face frontalization since it is firstly used by DR-GAN [33]. Later, TP-GAN [20] is proposed with a two-pathway structure and perceptual supervision. CAPG-GAN [17] introduce pose guidance through inserting conditional information carried by five-point heatmaps. PIM [37] aims to generate high-quality results through adding regularization items to learn face representations more robust to hard examples. CR-GAN [31] introduces a generation sideway to maintain the completeness of the learned embedding space and utilizes both labeled and unlabeled data to further enrich the embedding space for realistic generations. All those methods treat face frontalization as a 2D image-to-image translation problem without considering the intrinsic 3D properties of human face. They indeed perform well in the situation where training data is sufficient and captured well controlled. However, in-the-wild setting often leads to inferior performance, as we discussed in Sec. 1.

The attempt to combine prior knowledge of 3D face has been made by FF-GAN [35], 3D-PIM [38] and UV-GAN [7]. Their and our methods are all 3D-based but there are many differences. In FF-GAN, a CNN is trained to regress the 3DMM coefficients of the input. Those coefficients are integrated as a supplement of low-frequency information. 3D-PIM incorporates a simulator with the aid of a 3DMM to obtain prior information to accelerate the training process and reduce the amount of required training data. In contrast, we do not employ 3DMM to present shape or texture information. We introduce a novel dense correspondence field and frontalize the profiles through warping. UV-GAN leverages an out-of-the-box method to project a 2D face to a 3D surface space. Their network can be regarded as a 2D image-to-image translation model in the facial texture space. In contrast, once the training procedure is finished, our model can estimate the latent 3D information from the profiles without the need for any additional out-of-the-box methods.

It is also notable that some face frontalization methods tend to improve the performance of face recognition by data augmentation. Inspired by [30], DA-GAN [36], which acts as a 2D face image refiner, can be employed for pose-invariant face recognition. In brief, the refiner improves the quality of data augmented by ordinary methods. The training processes for face recognition methods benefit from these refined data and the performances are boosted. Thus, DA-GAN is a method for augmenting training data. Note that UV-GAN mentioned above can be used to benefit face recognition in the same manner with DA-GAN, so it is also a data augmentation method. In contrast, our HF-PIM is trained to directly rotate the given profile to the frontal face, which can be directly used for face recognition.

## 3 High Fidelity Pose Invariant Model

Given a profile face image $X$, our goal is to produce the frontal face image as close to the ground truth $Y$ as possible ($X, Y \in \mathbb{R}^{N \times N \times 3}$). As a reminder, we use $I_{ij}$ to denote the value of the pixel with coordinate $(i, j)$ in an image $I$. To learn the mapping, image pairs $(X, Y)$ are employed for model training. Inspired by recent progress in 3D face analysis [12, 13], we propose a brand-new framework which frontalizes given profile face through recovering geometry and texture information of the 3D face without explicitly building it. Concretely, the facial texture map and a novel dense correspondence field are leveraged to produce $Y$ through warping. The facial texture map $T$ lies in UV space - a space in which the manifold of the face is flattened into a contiguous 2D atlas. Thus, $T$ represents the surface of the 3D face. The dense correspondence field $F = (u; v)(u, v \in \mathbb{R}^{N \times N})$ is proposed to establish the connections between 2D and 3D surface spaces. $F$ is specified by the following statement: assuming that the coordinate of a point in $T$ is $(u_{ij}, v_{ij})$, the corresponding coordinate in $Y$ is $(i, j)$ after the warping operation. The right side on the top of Fig. 1 provides an intuitionistic illustration. Formally, given $T$ and $F$ with respect to $Y$, $T$ can be warped into $Y$ through the following formulation:

$$Y_{ij} = warp(i, j; \boldsymbol{F}, \boldsymbol{T}) = T_{u_{ij}, v_{ij}}, \ \ (i, j) \in \mathbb{F}, \tag{1}$$

where $\mathbb{F}$ is the coordinate set of those pixels standing for the facial part of $Y$. Our proposed warping procedure inherits the virtue of morphable model construction: geometry and texture are well disentangled whereas bound by dense correspondence. However, there are also some limitations, e.g., neglect of image background. Thus, additional process is necessary to produce the non-facial regions

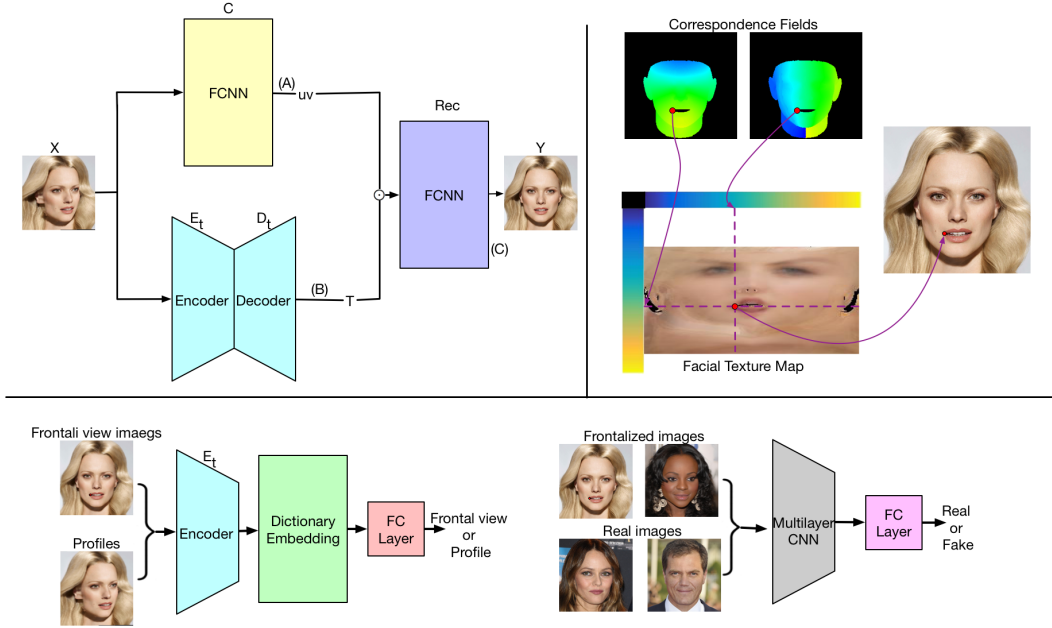

Figure 1: Left side on top: the framework of our HF-PIM to frontalize face images. The procedure consists of correspondence field estimation (A), facial texture feature map recovering (B) and frontal view warping (C). The right side on top: an illustration about the warping procedure discussed in Eq. 1. Those red dots and purple lines indicate the relationships between the facial texture map, correspondence field, and the RGB color image. Bottom side: the discriminators employed for ARDL (on the left) and ordinary adversarial learning (on the right).

for Eq. 1. To overcome these constraints, we employ a CNN-based wrapper $Rec$ to take $\boldsymbol{T}$ as well as $\boldsymbol{F}$ as the input and map them to $\boldsymbol{Y}$. Our $Rec$ can produce the non-fical parts simultaneously with the facial parts and keep the overall visual realism consistent with $\boldsymbol{Y}$. Concretely, $Rec$ is trained via optimizing the reconstruction loss:

$$L_{rec} = \|Rec(\boldsymbol{F}, \boldsymbol{T}) - \boldsymbol{Y}\|_1, \tag{2}$$

where $\|\cdot\|_1$ denotes calculating the mean of the element-wise absolute value summation of a matrix. Note that for our $Rec$, $\boldsymbol{T}$ is not limited to the RGB color space. In our experiment, we increase the number of feature channel to 32 and find that better performance is obtained.

In the following, we describe how to estimate the dense correspondence field $\boldsymbol{F}$ of the frontal view in Sec. 3.1. Then, the recovering procedure of the facial texture map $\boldsymbol{T}$ via ARDL is illustrated in Sec. 3.2. Regularization items and the overall loss function are introduced in Sec. 3.3.

## 3.1 Dense Correspondence Field Estimation

To obtain the ground truth dense correspondence field $\boldsymbol{F}$ of monocular frontal face images for training, we employ face reconstruction method for 3D shape estimation. Concretely, we employ BFM [28] as the 3D face model. Through the model fitting method proposed by [41], we get estimated shape parameters containing coordinates of vertices. To build $\boldsymbol{F}$, we map those vertices to UV space via the cylindrical unwrapping described in [3]. Those non-visible vertices are culled via z-buffering.

To infer the dense correspondence field of the frontal view from the profile image, we build a transformative autoencoder, $C$, with U-Net architecture. Given the input, $C$ first encodes it into pose-invariant shape representations and then recover dense correspondence field of frontal view. Further, those shortcuts in U-Net guarantee the preservation of spatial information in the output. To

supervise $C$ during training, we minimize the pixel-wise error between the estimated map and the ground truth $\boldsymbol{F}$, namely:

$$L_{corr} = \|C(\boldsymbol{X}) - \boldsymbol{F}\|_1. \tag{3}$$

### 3.2 Facial Texture Map Recovery

We employ a transformative autoencoder consisting of the encoder $E_t$ and the decoder $D_t$ for facial texture map recovering. However, the ground truth facial texture map $\boldsymbol{T}$ of monocular face image captured in the wild is absent. To sidestep the demand for $\boldsymbol{T}$, we introduce Adversarial Residual Dictionary Learning (ARDL) which provides supervision for learning. During the training procedure, only $\boldsymbol{Y}$ is required instead of $\boldsymbol{T}$.

The learning dictionary is set as: given a set of texture feature embeddings $\boldsymbol{B} = \{\boldsymbol{b}_1, \cdots, \boldsymbol{b}_n\}$ and a learnable codebook $\boldsymbol{C} = \{\boldsymbol{c}_1, \cdots, \boldsymbol{c}_m\}$ containing $m$ codewords with $d$ dimension, the corresponding residual vector is denoted as $\boldsymbol{r}_{ik} = \boldsymbol{b}_i - \boldsymbol{c}_k$, for $i = 1, \cdots, n$ and $k = 1, \cdots, m$. Through dictionary encoding, a fixed length representation $\boldsymbol{E} = \{\boldsymbol{e}_1, \cdots, \boldsymbol{e}_m\}$ can be calculated as follows:

$$\boldsymbol{e}_k = \sum_{i=1}^{n} \boldsymbol{e}_{ik} = \sum_{i=1}^{n} w_{ik}\boldsymbol{r}_{ik}, \tag{4}$$

where $w_{ik}$ is the corresponding weight for $\boldsymbol{r}_{ik}$. Inspired by [34] that assigns a descriptor to each codeword, we make those weights learnable. Concretely, the assigning weight is given by:

$$w_{ik} = \frac{\exp(-s_k\|\boldsymbol{r}_{ik}\|^2)}{\sum\limits_{j=1}^{m} \exp(-s_j\|\boldsymbol{r}_{ij}\|^2)}, \tag{5}$$

where $\boldsymbol{s} = (s_1, \cdots, s_m)$ is the smoothing factor, which is also learnable. We denote the mapping from feature embeddings to dictionary representation as $D_{dic}$. The encoder $E_t$ is also employed to extract features for the dictionary learning.

We combine dictionary representation with adversarial learning, i.e., propose ARDL based on such an observation: when the identity label is fixed, for $\boldsymbol{X}$ across different poses, the recovered texture map $\boldsymbol{T}$ should be invariant. To this end, $E_t$ should eliminate those discrepancies caused by different views and encode the input into pose-invariant facial texture representation. We introduce adversarial learning mechanism to supervise $E_t$ by making $D_{dic}$ as its rival. Formally, the adversarial loss introduced by ARDL is formulated as:

$$L_{adv} = \mathbb{E}_{\mathbf{X} \sim p_{data}}[\log D_{dic}(E_t(\mathbf{X}))]. \tag{6}$$

Accordingly, $D_{dic}$ is optimized to minimize:

$$L_{dic} = \mathbb{E}_{\mathbf{X},\mathbf{Y} \sim p_{data}}[\log D_{dic}(E_t(\mathbf{Y})) + \log(1 - D_{dic}(E_t(\mathbf{X}))], \tag{7}$$

where we add a fully connected (FC) layer upon $D_{dic}$ to make binary predictions standing for real and fake. Through optimizing Eq. 6 and 7 alternatively, $E_t$ manages to make the encodings of the profile and the frontal view as similar as possible. In the meantime, $D_{dic}$ tries to find the clues standing for pose information, which provides the adversarial supervision information for $E_t$.

### 3.3 Overall Training Method

Following previous work [21, 20], we also add the perceptual loss to integrate domain knowledge of identities. An identity preserving network, e.g., VGG-Face [26] or Light CNN [16], can be employed

to supervise the frontalized results to be as close to the ground truth as possible in feature-level. Formally, the perceptual loss is formulated as:

$$L_p = \|\phi(Rec(\boldsymbol{F}, \boldsymbol{T})) - \phi(\boldsymbol{Y})\|_2^2, \tag{8}$$

where $\phi(\cdot)$ denotes the extracted identity representation obtained by the second last fully connected layer within the identity preserving network and $\|\cdot\|_2$ denotes the vector 2-norm.

We also introduce the adversarial loss in the RGB color image space following those GAN-based methods [33, 20, 17, 37, 35]. A CNN named $D_{rgb}$ is employed to give adversarial supervision in color space. Note that our method can be easily extended to those advanced versions [1, 25] of GAN. But in this paper, we simply use the original form of adversarial loss function [10] to prove that the effectiveness comes from our own contributions.

In summary, all the involved algorithmic components in our network are differentiable. Hence, the parameters can be optimized in an end-to-end manner via gradient backpropagation. The whole training process is described in Algorithm 1.

---

**Algorithm 1** Training algorithm of HF-PIM

1: **Input:** profile $\boldsymbol{X}$, the ground truth frontal face $\boldsymbol{Y}$ with the ground truth dense correspondence field $\boldsymbol{F}$, maximum iteration $iter$ and the identity preserving network [16].
2: **Output:** the frontalized result $\hat{\boldsymbol{Y}}$
3: Initializing $C, E_t, D_t, Rec, D_{dic}, D_{rgb}$
4: $i \leftarrow 0$
5: **while** $i < iter$ **do**
6:      Sampling training data
7:      Model forward propagation
8:      Calculating $L_{rec}, L_{corr}, L_{dic}, L_{adv}$ and $L_p$
9:      Calculating the adversarial losses in the RGB color image space, i.e., $L_g$ (for the generator) and $L_d$ (for the discriminator)
10:      $L \leftarrow L_{rec} + L_{corr} + L_{adv} + L_p + L_g$
11:      Optimize $C, Rec, E_t, D_t$ by minimizing $L$
12:      Optimize $D_{dic}$ by minimizing $L_{dic}$
13:      Optimize $D_{rgb}$ by minimizing $L_d$
14:      $i \leftarrow i + 1$
15: **end while**

---

## 4 Experiments

### 4.1 Experimental Settings

**Datasets.** To demonstrate the superiority of our method in both controlled and unconstrained environments and produce high-resolution face frontalization results, we conduct our experiment on four datasets: Multi-PIE [11], LFW [18], IJB-A [23], and CelebA-HQ [22]. Multi-PIE is established for studying on PIE (pose, illumination and expression) invariant face recognition. 20 illumination conditions, 13 poses within 90 yaw angles and 6 expressions of 337 subjects were captured in controlled environments. LFW is a benchmark database for face recognition. Over 13,000 face images are captured in unconstrained environments. IJB-A is the most challenging unconstrained face recognition dataset at present. It has 5, 396 images and 20, 412 video frames of 500 subjects with large pose variations. CelebA [24] is a large-scale face attributes dataset. Contained images cover large pose variations and background clutter. CelebA-HQ is a high-resolution subset established by [22]. Since Multi-PIE, LFW and IJB-A consist of images with relatively low resolutions, we use CelebA-HQ for high-resolution ($256 \times 256$) face frontalization.

**Implementation Details.** The training set is drawn from Multi-PIE and CelebA-HQ. We follow the protocol in [33] to split the Multi-PIE dataset. The first 200 subjects are used for training and the rest 137 ones for testing. Each testing identity has one gallery image from his/her first appearance. Hence, there are 72,000 and 137 images in the probe and gallery sets, respectively. For CelebA-HQ, we apply head pose estimation [41] to find those frontal faces and employ them (19, 203 images) for

Table 1: Comparisons on rank-1 recognition rates (%) across views under Multi-PIE Setting 2.

| Method | $\pm15°$ | $\pm30°$ | $\pm45°$ | $\pm60°$ | $\pm75°$ | $\pm90°$ |
|---|---|---|---|---|---|---|
| DR-GAN [33] | 94.9 | 91.1 | 87.2 | 84.6 | - | - |
| FF-GAN [35] | 94.6 | 92.5 | 89.7 | 85.2 | 77.2 | 61.2 |
| Light CNN [16] | 98.6 | 97.4 | 92.1 | 62.1 | 24.2 | 5.5 |
| TP-GAN [20] | 98.7 | 98.1 | 95.4 | 87.7 | 77.4 | 64.6 |
| CAPG-GAN [17] | 99.8 | 99.6 | 97.3 | 90.3 | 83.1 | 66.1 |
| PIM [37] | 99.3 | 99.0 | 98.5 | 98.1 | 95.0 | 86.5 |
| **HF-PIM(Ours)** | **99.99** | **99.98** | **99.88** | **99.14** | **96.40** | **92.32** |

training. We choose those images with large poses (5, 998 ones) for testing. Apparently, there are no overlap between our training and testing sets. LFW and IJB-A are only used for testing. Note that the images selected for training in CelebA-HQ are all frontal view, and we employ the face profiling method in [40] to make corresponding profiles. We adapt the model architecture in [39] to build our networks. We use Adam optimizer with a learning rate of 1e-4 and $\beta_1 = 0.5, \beta_2 = 0.99$. Our proposed method is implemented based on the deep learning library Pytorch [27]. Two NVIDIA Titan X GPUs with 12GB GDDR5X RAM is employed for the training and testing process.

**Evaluation Metrics.** To measure the quality of frontalized faces, the most common method is to evaluate the face recognition/verification performances via "recognition via generation", which means profiles are frontalized first, and then the performance is evaluated on these processed face images. This evaluation manner prefers frontalization results that preserve more identity information and directly reflect the contributions of frontalization methods on face recognition. Thus, "recognition via generation" has been adopted by a series of existing methods [20, 17, 35, 37]. Besides, since photographic results also indicate the performances qualitatively, visual quality is also compared in our experiment, as most GAN-based methods do.

## 4.2 Frontalization Results in Controlled Situations

In this subsection, we systematically compare our method with DR-GAN, TP-GAN, FF-GAN, CAPG-GAN and PIM on the Multi-PIE dataset. Those profiles with extreme poses ($75°$ and $90°$) are very challenging cases. Our performances are tested following the protocol of the setting 2 provided by Multi-PIE. Remind that our performance is evaluated by the "recognition via generation" framework. Concretely, when evaluating on Multi-PIE, profiles are first frontalized by our model and then used directly for verification and recognition. As for evaluating on those in-the-wild datasets (discussed in the next subsection), all the faces are frontalized by our model since their yaw angles are not known in advance. After the frontalization preprocessing, Light CNN [16] is employed as the feature extractor. We compute the cosine distance of extracted feature vectors for verification and recognition. The results are reported across different poses in Table 1. Note that the manners for evaluating TP-GAN, FF-GAN, CAPG-GAN, and PIM are the same with our model. Light CNN is used for these methods except FF-GAN (their feature extractor is not publicly available). DR-GAN is evaluated in a different manner: the feature vectors are directly extracted by their model. Thus, no extra feature extractor is needed for DR-GAN. Besides frontalization methods, the performance of Light CNN is also included as the baseline. The results are reported across different poses in Table 1. For those poses less than $60°$, the performances of most methods are quite good whereas our method performs better. We infer that the performance has almost saturated in this case. For those extreme poses, our methods can still produce visually convincing results and achieve state-of-the-art recognition performance. In general, when testing on Multi-PIE, due to its balanced data distribution and highly controlled environment, most methods perform relatively well (except those extreme poses).

## 4.3 Frontalization Results in the Wild

Extending face frontalization to in-the-wild setting is a very challenging problem with significant importance. We focus on testing on IJB-A and LFW in this subsection. For LFW, we evaluate face verification performance on the frontalized results of the 6000 face pairs provided by the dataset. For IJB-A, both verification and identification are tested in 10-fold cross-validation. The results are

Table 2: Face recognition performance (%) comparisons for in-the-wild datasets. The left part is compared on LFW and the right side is on IJB-A. The results on IJB-A are averaged over 10 testing splits. "-" means the result is not reported.

| Method | LFW Verification | | Method | IJB-A Verification | | IJB-A Recognition | |
| | ACC | AUC | | FAR=0.01 | FAR=0.001 | Rank-1 | Rank-5 |
| --- | --- | --- | --- | --- | --- | --- | --- |
| TP-GAN [20] | 96.13 | 99.42 | DR-GAN [33] | 77.4±2.7 | 53.9±4.3 | 85.5±1.5 | 94.7±1.1 |
| FF-GAN [35] | 96.42 | 99.45 | FF-GAN [35] | 85.2±1.0 | 66.3±3.3 | 90.2±0.6 | 95.4±0.5 |
| Light CNN [16] | 99.39 | 99.87 | Light CNN [16] | 91.5±1.0 | 84.3±2.4 | 93.0±1.0 | - |
| CAPG-GAN [17] | 99.37 | 99.90 | PIM [37] | 93.3±1.1 | 87.5±1.8 | 94.4±1.1 | - |
| HF-PIM(Ours) | **99.41** | **99.92** | HF-PIM(Ours) | **95.2±0.7** | **89.7±1.4** | **96.1±0.5** | **97.9±0.2** |

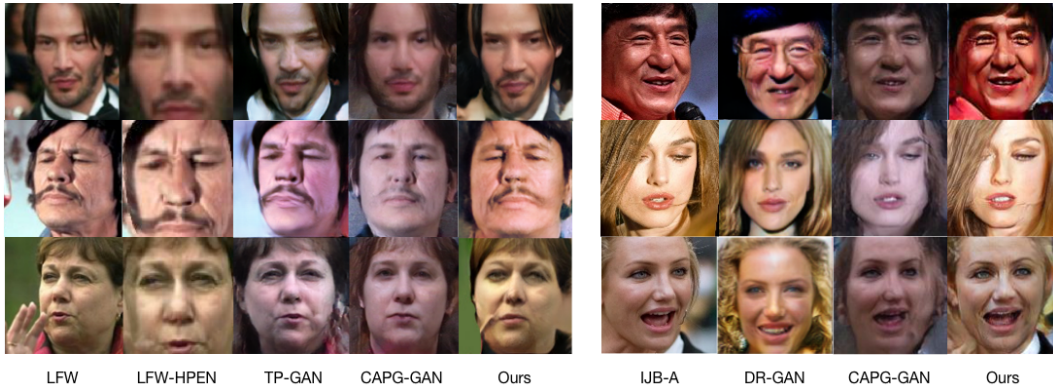

LFW     LFW-HPEN     TP-GAN     CAPG-GAN     Ours       IJB-A     DR-GAN     CAPG-GAN     Ours

Figure 2: Visual comparisons of face frontalization results. The samples on the left are drawn from LFW and the right side are from IJB-A.

summarized in Table 2. All the methods are tested with the same setting. Note that the training set of IJB-A is not been used by any involved method for comparison.

We can see that face frontalization methods only marginally improve the performance on LFW because most faces in this dataset are (near) frontal view. Besides, the baseline model Light CNN has already achieved a relatively high performance. But our method still outperforms existing frontalization methods in this case. When testing on IJB-A which contains lots of images with large and even extreme poses, our method shows a significant improvement for face verification and recognition. The visual comparison[2], which is shown in Fig 2, also proves our superiority of preserving identity information and texture details. Thanks to the 3D-based framework and powerful adversarial residual dictionary learning, our HF-PIM produces results with very high fidelity. For other methods, they indeed produce reasonable images but redundant manipulations can be observed. For instance, DR-GAN make the eyes of the subject in the middle in IJB-A open; TP-GAN and CAPR-GAN tend to change the skin color and background.

## 4.4 High-Resolution Face Frontalization

Generating high-resolution results has great importance on extending the application of face frontalization. However, due to its difficulty, few methods consider producing images with size larger than $128 \times 128$. To further demonstrate our superiority, frontalized $256 \times 256$ results on CelebA-HQ are proposed in this paper. Some samples are shown in Fig 3. We also make comparisons with TP-GAN and CAPG-GAN. Note that since results on CelebA-HQ have not been reported by previous methods, we contact the authors to get their model and produce $128 \times 128$ results through carefully following their instructions. The images in CelebA-HQ contain rich textures that are difficult for the generator to reproduce faithfully. Even in such a challenging situation, HF-PIM is still able to produce plausible results. The results of [17] and [20] look less appealing.

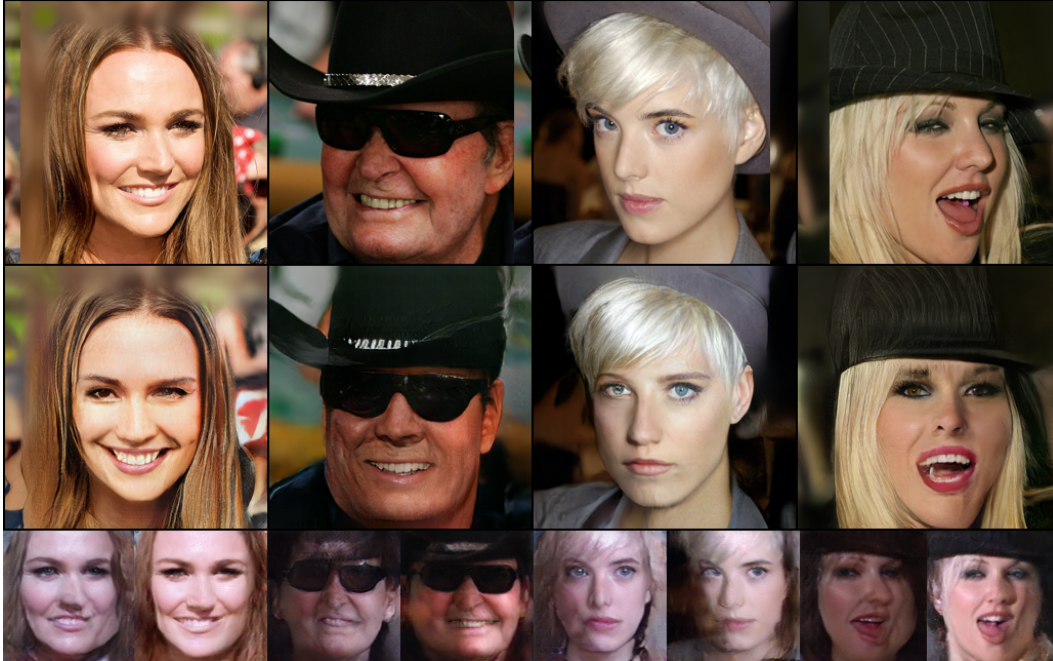

Figure 3: High-resolution frontalized results on the testing set of CelebA-HQ. The first row is the input profile images. The second row is the frontalized images produce by our HF-PIM. The results of CAPG-GAN (on the left for each subject) and TP-GAN (on the right) are shown in the third row.

Existing methods [20, 33, 37, 35, 17] measure the performance of face recognition to reflect the quality of frontalized results. This measurement cannot be applied to those datasets without identity labels (like CelebA-HQ) and neglects texture information that are not sensitive to identity. However, the neglected textures also play an import role on the visual quality and should be preserved faithfully. For face attribute analysis, data augmentation and many other practical applications, recovering high-resolution frontal view with detailed texture information has great potential for making progress. Finding new applications for face frontalization and putting forward new metrics need further research.

## 5   Conclusion

This paper has proposed High Fidelity Pose Invariant Model (HF-PIM) to produce realistic and identity-preserving frontalization results with a higher resolution. HF-PIM combines the advantages of 3D and GAN based methods and frontalizes profile images via a novel texture warping procedure. Through leveraging a novel dense correspondence field, the prerequisite of warping is decomposed into dense correspondence field estimation and facial texture map recovering, which are well addressed by a unified end-to-end deep network. We also have introduced Adversarial Residual Dictionary Learning (ARDL) to supervise facial texture map recovering without the need of 3D data. Exhaustive experiments have shown proposed method can preserve more identity information as well as texture details, which make the high-resolution results far more realistic.

## 6   Acknowledgments

This work is funded by the National Key Research and Development Program of China (Grant No. 2017YFC0821602, 2016YFB1001000) and the National Natural Science Foundation of China (Grant No. 61427811, 61573360).

## Footnotes

\*Ran He is the corresponding author.

[2]Visual results produced by other methods are released by their authors. Different methods usually report visual examples of different identities. We try our best to find those identities reported by most methods.

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
