[Supplementary Material]

# Learning a High Fidelity Pose Invariant Model for High-resolution Face Frontalization Supplementary Materials

**Jie Cao,  Yibo Hu,  Hongwen Zhang,  Ran He,** * **Zhenan Sun**
National Laboratory of Pattern Recognition, CASIA
Center for Research on Intelligent Perception and Computing, CASIA
Center for Excellence in Brain Science and Intelligence Technology, CASIA
University of Chinese Academy of Sciences, Beijing, 100049, China
{jie.cao,yibo.hu,hongwen.zhang}@cripac.ia.ac.cn   {rhe,znsun}@nlpr.ia.ac.cn

## 1   Implementation Details

**Network Architecture.** We adapt our network architectures from [5]. Below, we follow the naming convention of [2], Let $c7s1 - k$ denote a $7 \times 7$ Convolution-BatchNorm- ReLU layer with $k$ filters and stride 1. $dk$ denotes a $3 \times 3$ Convolution-BatchNorm-ReLU layer with $k$ filters, and stride 2. Reflection padding was used to reduce artifacts. $Rk$ denotes a residual block that contains two $3 \times 3$ convolutional layers with the same number of filters on both layer. $uk$ denotes a $3 \times 3$ fractional-strided-Convolution- BatchNorm-ReLU layer with $k$ filters, and stride $\frac{1}{2}$.

The encoder of $C$ consists of:

$$c7s1 - 32, d64, d128, R128 \times 6, u64, u32, c7s1 - 64$$

$E_t$ consists of:

$$c7s1 - 32, d64, d128, R128 \times 6, u64, u32, c7s1 - 3$$

Corresponding decoders have mirror structures of the encoders.

Our $Rec$ consists of:

$$c7s1 - 32, d64, d128, u64, u32, c7s1 - 3$$

**Weights for Different Losses.** The weight of $L_{rec}$ is assigned to 1. All the other weights are assigned to 0.1. We do not take much time on trying different weights and find this setting already produces very good results.

## 2   Face Profiling Method for Making Training Data Pairs

For the Multi-PIE dataset, subjects have images across different poses captured simultaneously. Those images are perfect for training. But images in CelebA-HQ are all captured in the wild. So, for the images in training set from CelebA-HQ, we first detect the landmarks through the method in [1]. We fit the 3DMM provide by [3] through the Multi-Features Framework [4] to get the estimated 3D shape information. Through 3D shape information, we rotate those images by employing the method in [6]. Some samples profiled in this procedure are shown in 1. Note that [6] has limited capacity for face frontalization because they cannot infer those missing texture. During training, we feed those profiles to our model as the input, and the frontal view images are used as the ground truth.

Figure 1: Examples of training data pairs. Those profiles are synthesized by other methods.

## 3 Visual Results on Multi-PIE

Frontalized examples on Multi-PIE are shown in Figures 2, 3, 4 and 5. For each subject, the input image is on the top, the frontalized result is in the middle and the ground truth for reference is on the bottom. The ID numbers in these figures are consecutive (from 201 to 300). We can see that for most subjects both the frontalized results preserve both the visual realism and the characteristics of identities very well. However, for instance, the results of subject 259 and 260 may look less convincing. We infer that it is caused by the unbalanced data distribution in race during the training procedure.

## 4 Additional Visual Results on IJB-A

More frontalized examples on IJB-A are shown in Fig. 6. The first and the third rows are the input images. The second and the fourth rows are corresponding frontalized images. Note that profiles drawn from in-the-wild datasets have no available frontal faces as the ground truth. We can see that for this challenging in-the-wild condition, the characteristics of identities are still well preserved by our HF-PIM.

## 5 Additional Visual Results on CelebA-HQ

For high-resolution frontalization, the flaws in texture recovery will be exaggerated. Besides, in-the-wild setting brings significant variations in pose, illumination and expression. Thus, face frontalization in CelebA-HQ is an extremely challenging task. We are the first to study on this case and obtain fairly visual appealing results, as shown in Fig. 7 and 8. For each subject, the input image is on the upper row and the frontalized result is on the lower row.

## Footnotes

*Ran He is the corresponding author.

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

Figure 7: More results on CelebA-HQ.

Figure 8: More results on CelebA-HQ.