[Reviews · NeurIPS 2018]

Reviewer 1



The paper proposes a face frontalization method based on a combination of existing methods of dense correspondence-based texture estimation and warping. An adversarial network was also trained in assisting the improve the texture map learning. The proposed method was evaluated via face recognition and verification tasks. Face frontalization is important specifically for applications in the wild with no pose and other conditional constraints. The topic is relevant to the NIPS audience. The paper reads good and is easy to follow. However, there are many weaknesses in the paper. I will list them as follows. Major comments: =============== - Since face recognition/verification methods are already performing well, the great motivation for face frontalization is for applications in the wild and difficult conditions such as surveillance images where pose, resolution, lighting conditions, etc… vary wildly. To this effect, the paper lacks sufficient motivation for these applications. - The major drawback is the method is a collection of many existing methods and as such it is hard to draw the major technical contribution of the paper. Although a list of contributions was provided at the end of the introduction none of them are convincing enough to set this paper aside technically. Most of the techniques combined are standard existing methods and/or models. If the combination of these existing methods was carefully analyzed and there were convincing results, it could be a good application paper. But, read on below for the remaining issues I see to consider it as application paper. - The loss functions are all standard L1 and L2 losses with the exception of the adversarial loss which is also a standard in training GANs. The rationale for the formulation of these losses is little or nonexistent. - The results for face verification/recognition were mostly less than 1% and even outperformed by as much as 7% on pose angles 75 and 90 (see Table 1). The exception dataset evaluated is IJB-A, in which the proposed model performed by as much as 2% and that is not surprising given IJB-A is collected in well constrained conditions unlike LFW. These points are not discussed really well. - The visual qualities of the generated images also has a significant flaw in which it tends to produce a more warped bulged regions (see Fig. 3) than the normal side. Although, the identity preservation is better than other methods, the distortions are significant. This lack of symmetry is interesting given the dense correspondence is estimated. - Moreover, the lack of ablation analysis (in the main paper) makes it very difficult to pinpoint from which component the small performance gain is coming from. - In conclusion, due to the collection of so many existing methods to constitute the proposed methods and its lack of convincing results, the computational implications do not seem warranted. Minor comments: =============== - Minor grammatical issue need to be checked here and there. - The use of the symbol \hat for the ground truth is technically not appealing. Ground truth variables are usually represented by normal symbols while estimated variables are represented with \hat. This needs to be corrected throughout for clarity.

Reviewer 2



This submission proposes a High Fidelity Pose Invariant Model (HF-PIM) that combines the advantages of 3D and GAN based methods to frontalize profile faces. Pros: -- A novel High Fidelity Pose Invariant Model (HF-PIM) is proposed to produce more realistic and identity-preserving frontalized face images with a higher resolution. -- A method named ARDL is proposed to supervise the process of facial texture map recovering without using any 3D information. Cons: -- Section 3.2 is a little hard to follow. If possible, please make this section more clear and easier to understood in the modified submission. -- The authors claimed that "Exhaustive experiments on both controlled and uncontrolled environments demonstrate that the proposed method not only boosts the performance of pose-invariant face recognition but also dramatically improves high-resolution frontalization appearances." But in Table I, the mean accuracy of HF-PIM(Ours) seems much lower than the one of PIM. -- The results (generated faces) of proposed method in Fig. 3 seems not symmetric. Is it possible to solve this problem?

Reviewer 3



This paper presents a method to produce realistic and identity-preserving frontalized face images with a high resolution. The proposed method combines the advantages of 3D and GAN based methods and frontalizes profile images via a novel facial texture fusion warping procedure. Through leveraging a novel dense correspondence field, the prerequisite of warping is decomposed into correspondence field estimation and facial texture recovering, which are well addressed by a unified deep network in an end-to-end manner. The authors also introduced Adversarial Residual Dictionary Learning to supervise facial texture map recovering without the need of 3D data. Thorough experimental results on Multi-PIE, LFW, IJB-A, and CelebA-HQ demonstrate that the proposed method not only boosts pose-invariant face recognition in the wild, but also dramatically improves the visual quality of high-resolution images. I believe this paper shows a promising approach to solve the face frontalization issue for pose-invariant face recognition that I have not seen elsewhere so far. The paper is written clearly, the math is well laid out and the English is fine. I think it makes a clear contribution to the field and can promote more innovative ideas, and should be accepted.